

# Seasonal modeling analysis of nitrate formation pathways in Yangtze River Delta region, China

Jinjin Sun[1,2], Momei Qin[1*], Xiaodong Xie[1], Wenxing Fu[1], Yang Qin[1,2], Li Sheng[1], Lin Li[1], Jingyi Li[1], Ishaq Dimeji Sulaymon[1], Lei Jiang[1], Lin Huang[1], Xingna Yu[2], Jianlin

Hu[1*]

[1] Jiangsu Key Laboratory of Atmospheric Environment Monitoring and Pollution Control, Collaborative Innovation Center of Atmospheric Environment and Equipment Technology, Nanjing University of Information Science & Technology, Nanjing 210044,

China

[2] Key Laboratory of Meteorological Disaster, Ministry of Education, Joint International Research Laboratory of Climate and Environment Change, Collaborative Innovation Center on Forecast and Evaluation of Meteorological Disasters, Key Laboratory for Aerosol-Cloud-Precipitation of China Meteorological Administration, Nanjing

University of Information Science and Technology, Nanjing 210044, China

*Correspondence to:* Jianlin Hu (jianlinhu@nuist.edu.cn) and Momei Qin (momei.qin@nuist.edu.cn)





## Abstract

Nitrate ($NO_3^-$) has been the dominant and the least reduced chemical component of fine particulate matter ($PM_{2.5}$) since the stringent emission control implemented in China in 2013. The formation pathways of $NO_3^-$ vary seasonally and differ substantially in daytime vs. nighttime. They are affected by precursor emissions, atmospheric oxidation capacity, and meteorological conditions. Understanding $NO_3^-$ formation pathways provides insights for the design of effective emission control strategies to mitigate $NO_3^-$ pollution. In this study, the Community Multiscale Air Quality (CMAQ) model was applied to investigate the impact of regional transport, predominant physical processes, and different formation pathways to $NO_3^-$ and total nitrate ($TNO_3$, i.e., $HNO_3 + NO_3^-$) production in the Yangtze River Delta (YRD) region during the four seasons of 2017. $NO_3^-/PM_{2.5}$ and $NO_3^-/TNO_3$ are the highest in the winter, reaching 21% and 94%, respectively. Adjusted gas ratio (adjGR = $([NH_3] + [NO_3^-])/([HNO_3] + [NO_3^-])$) in YRD is generally greater than two in different seasons across most areas in YRD, indicating that YRD is mostly in the $NH_3$-rich regime and $NO_3^-$ is limited by $HNO_3$ formation. Local emissions and regional transportation contribute to YRD $NO_3^-$ concentrations by 50–62% and 38–50%, respectively. Majority of the regional transport of $NO_3^-$ concentrations is contributed by indirect transport (i.e., $NO_3^-$ formed by transported precursors reacting with local precursors). Aerosol (AERO, including condensation, coagulation, new particle formation and aerosol growth) processes are the dominant source of $NO_3^-$ formation. In summer, $NO_3^-$ formation is dominated by AERO and total transport (TRAN, sum of horizontal and vertical transport) processes. The OH+$NO_2$ pathway contributes to 60–83% of the $TNO_3$ production, and the $N_2O_5$ heterogeneous (HET $N_2O_5$) pathway contributes to 10–36% in YRD. HET $N_2O_5$ contribution becomes more important in cold seasons than warm seasons. Within the planetary boundary layer in Shanghai, the $TNO_3$ production is dominated by the OH+$NO_2$ pathway during the day (98%) in the summer and spring, and by the HET



N₂O₅ pathway during the night (61%) in the winter. Local contribution dominates the OH+NO₂ pathway for TNO₃ production during the day, while indirect transport dominates the HET N₂O₅ pathway at night.

**Keywords:** Nitrate formation pathways; chemical transport model, process analysis; local and transport contributions; Yangtze River Delta.

## 1. Introduction

The Yangtze River Delta (YRD) region, located in eastern China, is among the most populous and developed economic regions in China. Because of rapid population growth, economic advancement, urbanization, and industrialization during recent decades, the YRD region has been frequently suffering from both fine particulate matter (PM$_{2.5}$) and ozone (O$_3$) pollution problems (Qin et al., 2021;Sun et al., 2019;Dai et al., 2021). Particulate nitrate (NO$_3^-$) is a major PM$_{2.5}$ component and high concentrations of NO$_3^-$ are often observed during cold seasons in the YRD region, due to high precursors emissions and regional transport contribution. Huang et al. (2014) reported that the daily average PM$_{2.5}$ concentrations in Shanghai were 91 μg m$^{-3}$ during haze pollution events of 5–25 January 2013, whereas NO$_3^-$ accounted for 14% total PM$_{2.5}$ mass. Huang et al. (2020a) observed that PM$_{2.5}$ concentrations in Nanjing were 271 μg m$^{-3}$ on 30–31 December of 2017, and the fraction of NO$_3^-$ was ~27%. Lin et al. (2020) found that the peak concentration of NO$_3^-$ in Nanjing was 85 μg m$^{-3}$ during haze pollution events in the spring of 2016.

Owing to the stringent emission control strategies implemented in China since 2013, PM$_{2.5}$, sulfur dioxide (SO$_2$) and nitrogen oxides (NO$_x$ = nitric oxide (NO) + nitrogen dioxide (NO$_2$)) emissions have decreased substantially, which led to significant decreases in primary PM$_{2.5}$ and sulfate (SO$_4^{2-}$) concentrations in China (Li et al., 2022;Chen et al., 2021). However, compared to SO$_4^{2-}$ and other PM$_{2.5}$



components, the reduction rate of $NO_3^-$ was much less slower (Wen et al., 2018;Zhai

et al., 2021;Zhou et al., 2022;Wang et al., 2022). This led to a rise in the ratio of

$NO_3^-$ mass to total $PM_{2.5}$ in eastern China, rendering $NO_3^-$ the dominant chemical

component of $PM_{2.5}$ (accounting for 24–35 %, especially during the cold season and

haze pollution events) (Ding et al., 2019;Wen et al., 2018;Lin et al., 2020;Fu et al.,

2020;Zhou et al., 2022). High concentrations of $NO_3^-$ influence the hygroscopicity

and optical properties of particles, contributing to the formation of haze and to

visibility degradation (Hu et al., 2021;Xie et al., 2020). Mitigating $NO_3^-$ pollution has

become an urgent concern in YRD.

$NO_3^-$ is formed in the atmosphere by a series of chemical reactions leading to the

production of nitric acid ($HNO_3$) and then following gas-to-particle partitioning

(Griffith et al., 2015;Guo et al., 2018;Lin et al., 2020). The key $NO_3^-$ formation

pathways include the gas-phase oxidation (hydroxyl (OH) and $NO_2$) and the

heterogeneous hydrolysis of dinitrogen (HET $N_2O_5$) on the wet particles' surface (Fan

et al., 2021;Wang et al., 2018;Chen et al., 2020). Several studies investigated the

importance of different pathways to $NO_3^-$ formation in various locations using

chemical transport models (CTMs), field observations, the box model, or oxygen and

nitrogen isotope techniques. For example, He et al. (2020) and Li et al. (2021b)

reported that the OH+$NO_2$ pathway dominates daytime $NO_3^-$ formation in the YRD,

accounting for 60–92 % and 55–86 % in warm and cold seasons, respectively. The

HET $N_2O_5$ pathway is the main nocturnal-$NO_3^-$ formation in winter, especially in

severe haze episodes, with contributions of 44–97 % at night (Fu et al., 2020;He et al.,

2018). Furthermore, Tan et al. (2021) and Wang et al. (2018) indicated that the

chemical formation cannot explain the variation of $TNO_3$ at the surface (sum of $NO_3^-$

and $HNO_3$), due to the concentrations of $N_2O_5$ being close to zero and controlled by

high NO emissions at night. Fan et al. (2021) and Kim et al. (2014) further emphasized

the contributions of $NO_3^-$ formation pathways differ significantly at vertical altitudes,



owing to the vertical gradients of nocturnal $NO_3$ and total oxidant ($NO_2+O_3$) level within the planetary boundary layer (PBL). Prabhakar et al. (2017) revealed that the active nocturnal $NO_3^-$ formation via the HET $N_2O_5$ pathway from the upper PBL contributed to daytime surface $NO_3^-$ concentrations in California, accounting for 80 %.

The complex $NO_3^-$ formation chemistry involves the anthropogenic emission of precursors (i.e., $NO_x$, and ammonia ($NH_3$)) and atmospheric oxidants (i.e., OH, $O_3$, and $N_2O_5$) (Chan et al., 2021;Womack et al., 2019). Studies suggested that $NO_3^-$ responds nonlinearly to its precursors emissions reductions in major Chinese regions (i.e., the North China Plain (NCP) and YRD), emphasizing that the uncoordinated

control of precursors (i.e., $SO_2$, $NH_3$, and $NO_x$) increase the atmospheric oxidant capacity (AOC) and enhance $NO_3^-$ formation in $NO_x$-rich regimes (Li et al., 2021b;Huang et al., 2020b;Lu et al., 2021a). Coupled with the chemical formation, regional transport also plays important roles in $NO_3^-$ pollution formation. Previous modeling studies using the CTMs highlighted the important role of the regional

transport in $NO_3^-$ concentrations in major regions of eastern China (Itahashi et al., 2017;Qu et al., 2021;Ying et al., 2014;Shen et al., 2020). For example, Huang et al. (2020a) reported that secondary pollutants are regionally transported between the NCP and YRD regions (a distance of 1000 km), and hence simultaneously exacerbate the levels of secondary inorganic aerosols (SIA) in two major Chinese regions. Ying et al.

(2014) revealed that the regional air pollution transport from the north and central China contributed about 45 % to $NO_3^-$ in Shanghai during the winter of 2009. Wu et al. (2017) suggested that the regional transport plays a key role in $NO_3^-$ sources in Shanghai (accounting for about 90 %), while local emission only contributed 10 % for $NO_3^-$ in January 2013. Shen et al. (2020) reported that the contribution of regional

transport amounted to around 60–98 % to the high concentrations of $NO_3^-$ under severe haze episodes in two winters of 2015 and 2016 in the YRD. Qu et al. (2021) found that the indirect transport made a contribution of 43 % to $NO_3^-$ in the Pearl River Delta





(PRD) region in cold season of 2015, mainly due to chemical reactions between the locally emitted NOx and transported $O_3$ at night. Du et al. (2020) also revealed that regional transport contributed about 56 % to $NO_3^-$ in Beijing in winter 2017, mainly produced via indirect transport.

The $NO_3^-$ formation pathways and controlling factors can be very different in different seasons even in the same studying region. Most previous studies on $NO_3^-$ focused on one or a few short period of pollution episodes, but lack of seasonal analysis. This study aims to obtain a comprehensive understanding of the seasonal variations in the $NO_3^-$ formation mechanisms, as well as to determine key precursors, dominant processes and chemical pathways in YRD. The Community Multiscale Air Quality (CMAQ) model was employed to investigate the contributions of various physical and chemical processes to $NO_3^-$ and $HNO_3$ formation. Regional transport and chemical reaction pathways were quantified for the YRD region. The analyses were conducted in the four seasons of 2017 to compare and identify the key impact factors for $NO_3^-$ in different seasons, and to provide a scientific basis for designing effective emissions control strategies to mitigate the urgent $NO_3^-$ pollution in the YRD region.

## *2. Methods*

### *2.1. Model configuration*

The CMAQ v5.2 model (Wyat Appel et al., 2018;Liu et al., 2020b;Sheng et al., 2022) was applied to investigate the major chemical pathways and physical processes that contribute to $NO_3^-$ and $TNO_3$ formation in the YRD region. Two nested domains were used, as shown in Fig. 1. The outer domain (36 km horizontal resolution) spanned eastern and southeastern China, while the inner domain (12 km horizontal resolution) spanned the entire YRD region. The simulation periods were January, April, July, and October 2017, representing the winter, spring, summer, and autumn, respectively. The simulation began three days prior to each of the study periods, and the results were not

included in the model analysis as they served as a spin-up of the model.

The CMAQ model was configured using the photochemical mechanism of the State-wide Air Pollution Research Center version 07 (SAPRC07tic) and the sixth-generation aerosol (AERO6i) module (Fu et al., 2020;Sulaymon et al., 2021). Further details about the CMAQ modeling system provided in previous studies (Hu et al., 2016;Liu et al., 2020b). The Weather Research and Forecasting model (WRF v4.2,

http://www.wrf-model.org) was used to simulate the required meteorological fields inputs, with initial and boundary meteorological conditions from the 1°×1° National Centers for Environmental Prediction Final (NCEP/FNL) reanalysis data (https://rda.ucar.edu/datasets/ds083.2/). Detailed configurations of the WRF model provided in the studies of Hu et al. (2016) and Wang et al. (2021), shown in Table S1,.

The anthropogenic emissions of 2017 in the YRD region were released by the Shanghai Academy of Environmental Sciences (SAES) (An et al., 2021), and used for the entire YRD region. The Multi-resolution Emission Inventory for China of the year 2017 with resolution of 0.25° × 0.25° (MEIC v1.3, http://meicmodel.org) served as the anthropogenic emissions for other Chinese regions outside the YRD (Zheng et al.,

2018). Emissions from other regions outside China in the inner domain were calculated using the gridded Regional Emission inventory in ASia (REAS v3.2, 0.25° × 0.25° resolution) emissions of the year 2015. The global model of emissions of gases and aerosols from nature (MEGAN v2.1) was used to estimate biogenic emissions (Guenther et al., 2012). Biomass burning emissions were based on satellite observations

including both gases and aerosols from the 2017 Fire Inventory from NCAR (FINN) (Wiedinmyer et al., 2011). Further descriptions of the emissions processing are provided in previous studies by Hu et al. (2016) and Qiao et al. (2015), and therefore not repeated here.

### 2.2. Contributions of transport

To quantify the contributions of local and regional transport to the surface

concentrations of the nitrate-phase species (i.e., $HNO_3$ and $NO_3^-$), four scenarios were simulated under the same meteorological fields. Briefly, in the first (base) scenario, the anthropogenic emissions of 2017 in the YRD and outside regions were included. In the second (YRD-zero) scenario, anthropogenic emissions in the YRD were set to zero, while anthropogenic emissions in regions outside YRD were used. In the third (outside-zero) scenario, only anthropogenic emissions in the YRD were included, while the regions outside the YRD were set to zero. The fourth (all-zero) scenario represented the background case, where the anthropogenic emissions within the study domain were set to zero.

The predicted concentrations were denoted as $C_{base}$, $C_{YRD-zero}$, $C_{outside-zero}$, and $C_{all-zero}$, representing concentration associated with the base, YRD-zero, outside-zero, and all-zero scenarios, respectively. For $NO_3^-$ in YRD, the contributions of local YRD emissions, direct transport ($NO_3^-$ contributed by transported precursors from outside regions), indirect transport ($NO_3^-$ contributed by transported and local-emitted precursors), and background were defined as $F_{Local}$, $F_{Direct}$, $F_{Indirect}$, and $F_{Background}$, and they were quantified as follows:

$$F_{Local} = (C_{outside-zero} - C_{all-zero})/C_{base} \qquad (1)$$

$$F_{Direct} = (C_{YRD-zero} - C_{all-zero})/C_{base} \qquad (2)$$

$$F_{Indirect} = [(C_{base} - C_{outside-zero}) - (C_{YRD-zero} - C_{all-zero})]/C_{base} \qquad (3)$$

$$F_{Background} = C_{all-zero}/C_{base} \qquad (4)$$

In addition to $NO_3^-$, the major gases (i.e., $O_3$, $NH_3$, $NO_2$, and $HNO_3$), atmospheric oxidants (i.e., OH, and $N_2O_5$) and particulate pollutants ($PM_{2.5}$, $SO_4^{2-}$, and SOC) were also quantified. The values of the contributions of the local, direct and indirect transport emissions can be greater or less than zero, which represents the generation or depletion of pollutants through chemical reactions between local and non-local precursors.

### 2.3. Process analysis

In the CMAQ model system, the process analysis (PA) tool has two components,



including the Integrated Process Rate (IPR) and Integrated Reaction Rate (IRR) (Liu et al., 2011;Byun and Schere, 2006). The IPR analysis was applied to investigate the

cumulative effect of chemical and physical processes to $NO_3^-$ and $HNO_3$ formation and their daily variation within the PBL (Chen et al., 2019;Yang et al., 2020;Kim et al., 2014). These processes, as explained in Table S2, include aerosol processes (AERO), gas chemistry (CHEM), emission (EMIS), horizontal transport (HTRA), vertical transport (VTRA), dry deposition (DDEP), and cloud processes (CLDS). Furthermore,

the IRR analysis was employed to quantify the rates of $TNO_3$ chemical reactions pathways (Qu et al., 2021;Fu et al., 2020;Shen et al., 2020). The complex chemical production of $TNO_3$ involves eight reactions pathways, detailed in Table S3 (Qu et al., 2021;Fu et al., 2020;Chuang et al., 2022). In the latter analyses, these pathways are grouped into three major $TNO_3$ production pathways, including the $OH+NO_2$, HET

$N_2O_5$, and "Others" pathways, according to their importance. Shanghai is selected as an example in the IPR and IRR analysis to explore the impacts of physical and chemical processes of $NO_3^-$ and $HNO_3$ formation because it is the largest city in YRD and has the most abundant measurement data.

*2.4. Observation data*

Hourly concentrations of six routine air pollutants (i.e., $O_3$, $PM_{2.5}$, $NO_2$, $SO_2$, and carbon monoxide (CO)) in five representative YRD cities (i.e., Shanghai, Nanjing, Hefei, Hangzhou, and Changzhou, shown in Fig. 1) during the four seasons were obtained from the China Ministry of Ecology and Environment (http://106.37.208.233:20035/, last accessed on April 30, 2022). Furthermore, hourly

$NO_3^-$ concentrations were measured by the Monitors for AeRosols and Gases (MARGA 1S ADI 2080, Netherlands) (Khezri et al., 2013) at four urban atmospheric environment supersites, including Shanghai (31.23°N, 121.54°E), Hefei (31.78°N, 117.20°E), Hangzhou (30.29°N, 120.16°E), and Changzhou (31.76°N, 119.96°E). Observation data of meteorological parameters (temperature (T2, °C), relative humidity (RH, %),



wind speed (WS, m/s) and wind direction (WD, °)) for 75 weather stations in YRD were downloaded from the Chinese Meteorological Agency (http://data.cma.cn/en, last accessed on November 30, 2021).

     The statistical metrics used for the WRF-CMAQ model evaluation include the mean bias (MB), normalized mean bias (NMB), normalized mean error (NME),

correlation coefficient (R), root mean square error (RMSE), and index of agreement (IOA). Definitions and criteria of all statistical metrics are illustrated in Table S4. The benchmarks of major air pollutants ($PM_{2.5}$, $NO_2$, $O_3$, and $NO_3^-$) concentrations are suggested by Emery et al. (2017) and Huang et al. (2021). The benchmarks of major meteorological parameters (T2, WS, and WD) are suggested by Emery and Tai (2001).

## 3. Results and discussion

### 3.1. Model evaluation

#### 3.1.1. WRF model performance

     Table 1 shows the modeling performance statistics of the meteorological parameters in the four seasons of 2017. Predicted T2 and WS values are slightly higher than the observations, and MB values of T2 and WS exceed the suggested benchmark

(MB ≤ ±0.5) in all seasons. The seasonal and annual IOA values of T2 occur within the suggested benchmark (IOA ≥ 0.8). For WS, the seasonal and annual values of RMSE and IOA all meet the suggested criterion (RMSE ≤ 2.0 and IOA ≥ 0.6). The MB values of WD are slightly above the suggested benchmark (MB ≤ ±10) in the four seasons,

except during spring. RH is generally under-estimated compared to the observations with averaged MB values of –6.96, –10.7, –9.06, and –5.98 in winter, spring, summer, and autumn, respectively. No suggested criterion for MB value of RH. In addition, high seasonal and annual values of R (0.85–0.95 for T; 0.87–0.91 for RH; 0.70–0.85 for WS; and 0.75–0.89 for WD) are found. The WRF performance in this study is comparable

to WRF performance in previous studies (Wang et al., 2021;Hu et al., 2016;Sulaymon



et al., 2021).

*3.1.2. CMAQ model performance*

Table 2 and Fig. S1 show the model performance and time series of major air pollutants in the four seasons. Overall, the CMAQ model has reasonably reproduced the observed $PM_{2.5}$, $O_3$, and $NO_2$ concentrations in the YRD region, especially in Shanghai. The daily concentrations of $PM_{2.5}$ are efficiently simulated in the five cities except Hefei, illustrated by the NMB, NME, and R values meeting the criteria established by Emery et al. (2017) (NMB ≤ ±0.30, NME ≤ 0.50, and R > 0.70). MDA8 $O_3$ are slightly overestimated in Nanjing, Hefei, Hangzhou, and Changzhou. Predicted concentrations of $NO_2$ are generally lower than the observations in all five cities (–0.15 < NMB ≤ –0.05, –10.37 < MB ≤ –1.89). When compared to previous air quality simulation studies (Hu et al., 2016;Wang et al., 2021;Ma et al., 2021;Sulaymon et al., 2021;Li et al., 2021a), the results in this study show a better model performance.

Fig. 2 illustrates the comparison of predicted and observed $NO_3^-$ concentrations at the four supersites on daily timescales (Fig. S2 shows the hourly predicted and observed $NO_3^-$ concentrations). The general temporal variations of observed $NO_3^-$ concentrations are efficiently captured by the model. Good agreement between predicted and observed values is demonstrated on daily timescales, especially in Shanghai (NMB = –0.49, R = 0.70), Hangzhou (NMB = 0.11, MB = 0.64) and Changzhou (NMB = 0.36, R = 0.56). The seasonal daily concentrations of $NO_3^-$ are efficiently predicted in Shanghai, Hangzhou and Changzhou, within the benchmark (NMB ≤ ±0.60, NME ≤ 0.75, and R > 0.6). The performance statistical metrics of predicted $NO_3^-$ in this study are comparable to those of previous studies (Shi et al., 2017;Qu et al., 2021). $NO_3^-$ concentrations are generally underestimated during the summer and autumn. One possible reason is that RH is slightly underestimated by the WRF model during these seasons (Table 1), which results in a lower buildup of $NO_3^-$ concentrations. Other reasons could be associated with uncertainties in the $NO_3^-$ formation mechanisms (i.e., missing or insufficient



heterogeneous reactions in the current CMAQ model) and uncertainties in NOx and NH₃ emissions (Zheng et al., 2020;Lu et al., 2021b;Zheng et al., 2015;Liu et al., 2019).

### 3.2. Regional transport contribution to nitrate in YRD

Fig. S3 shows the spatial distribution of the seasonal (winter, spring, summer and autumn) and annual mean (average of the four seasons) $NO_3^-$, $HNO_3$, and $TNO_3$ concentrations under four different emissions scenarios in the d02 domain. Under $C_{base}$, the seasonal and annual predicted concentrations of $NO_3^-$ for the entire YRD region were 16.0, 7.4, 1.0, 5.4, and 7.4 μg m⁻³, respectively (Table S5). Compared to $C_{base}$, the seasonal and annual YRD $NO_3^-$ concentrations in $C_{outside-zero}$ decreased by 8.0, 2.8, 0.4, 2.2, and 3.3 μg m⁻³, respectively. Even more significant differences in $NO_3^-$ are observed between $C_{base}$ and $C_{YRD-zero}$. The $NO_3^-$ values decreased by 12.0, 6.9, 0.9, 4.8 and 6.1 μg m⁻³ in winter, spring, summer, autumn, and the year respectively, to become almost twice as high as those between $C_{base}$ and $C_{outside-zero}$. The results suggest that the YRD local anthropogenic emissions contribute more to the seasonal $NO_3^-$ concentrations.

Fig. 3 shows the regional contributions of the background, local, direct and indirect transport to nitrate-related species in the four seasons (results for Shanghai are shown in Fig S5). The local emissions dominate YRD $NO_3^-$, accounting for 50.4–62.0 % in the four seasons (Fig. 3a). Fig 3c suggests that the precursors ($NO_2$ and $NH_3$) are dominated by the local emissions (more than 93.0%). The contributions of the total regional transport (sum of indirect and direct transport) are 49.5, 38.0, 41.6, and 42.0 % in winter, spring, summer, and autumn, respectively. The indirect transport contributes 24.2–37.0% of $NO_3^-$ concentrations, and exceeds the contributions from direct transport in the spring, summer, and autumn. Similarly, Qu et al. (2021) reported that the reaction between the locally emitted NOx and transported $O_3$ dominates the production of indirect $NO_3^-$ transport in the PRD region.

In Fig. 3b, the local emission and indirect transport have negative contributions to $O_3$ concentration, leading to the depletion of $O_3$ in the four seasons. The local emissions





have negative contribution to $O_3$ in winter (–45.6%) and autumn (–12.3%), and the

indirect transport has negative contribution in spring (–8.5%) and autumn (–8.7%). For

$O_3$ and OH (Fig. 3b and 3d), indirect transport contributes about –8% and –12% – –42%

in all seasons, respectively. The negative contributions to $O_3$, $N_2O_5$, and OH suggest that

the atmospheric oxidants are consumed in YRD, which in turn enhances the chemical

production of $NO_3^-$.

### 3.3. Formation processes of nitrate

Fig. 4 shows the modeled diurnal variations of three nitrate-phases ($NO_3^-$, $HNO_3$,

and $TNO_3$), the major precursors (i.e., $O_3$, $NO_2$, and $NH_3$), and the major atmospheric

oxidants (OH and $N_2O_5$) in the four seasons for the entire YRD region in the base

scenario. Except for summer, higher predicted $TNO_3$ and $NO_3^-$ concentrations are

observed in early morning hours (6:00–8:00 am), while lower $TNO_3$ and $NO_3^-$

concentrations are observed around 16:00–18:00 pm. Predicted concentrations of $TNO_3$,

$HNO_3$, and $O_3$ show the same diurnal variations in the summer, and peak around 12:00

pm (the most active photochemical hours). The opposite profiles of $TNO_3$'s diurnal

variation between summer and non-summer are mainly attributed to the temperature

effect on the gas-to-particle partitioning between $NO_3^-$ and $HNO_3$. As shown in Fig. S3,

$NO_3^-$ dominates the $TNO_3$ concentrations and determines its diurnal variations in non-

summer, while $HNO_3$ dominates the diurnal variation in summer. A two-peak mode

diurnal variation of $NO_2$ and $NH_3$ is identified in the four seasons. High concentrations

of $NO_2$ and $NH_3$ occur in the early morning (hours 6:00–8:00 am) and early evening

(hours 18:00-19:00 pm), due to the local transportation emissions during rush hours.

OH and $N_2O_5$ have a one-peak mode diurnal variation in the four seasons. OH peaks

around 12:00 pm, similar to $HNO_3$, while $N_2O_5$ peaks around 18:00–20:00 pm.

Fig. S7 shows seasonal variations in $NO_3^-$/$PM_{2.5}$, $NO_3^-$/$TNO_3$, nitrogen oxidation

ratios (NOR = $[NO_3^-]$/($[NO_3^-] + [NO_2]$)), and adjusted gas ratio (adjGR = ($[NH_3]$ +

$[NO_3^-]$)/($[HNO_3] + [NO_3^-]$)) in YRD. $NO_3^-$/$PM_{2.5}$ and $NO_3^-$/$TNO_3$ are the highest in the



winter, accounting for 21 ± 5% and 94 ± 3%, respectively. The averaged NOR values for the entire YRD region are 0.24, 0.16, 0.03, and 0.13 mol/mol in winter, spring, summer, and autumn, respectively. The highest value of NOR in winter suggests a high conversion efficiency of $NO_2$ to $NO_3^-$. AdjGR values are generally greater than two in the four seasons across most areas in YRD, indicating that YRD is mostly in the $NH_3$-rich regime. Therefore, $NH_3$ is not a limiting factor of $NO_3^-$ formation in YRD.

Fig. 5 illustrates a two-peak mode diurnal variation of the net IPRs rates of $NO_3^-$ production in the four seasons. Peak hours are around mid-noon (10:00–11:00 am) and early evening (19:00–21:00 pm), with peak rates of 1.2–1.5, 0.7–0.8, 0.4–0.6, and 0.1–0.2 $\mu$g m$^{-3}$ h$^{-1}$ in the winter, spring, summer, and autumn, respectively. AERO processes (including condensation, coagulation, and aerosol growth) are the dominant contributors of $NO_3^-$ formation, with the peak rates of 2.1, 1.3, 1.5, and 0.4 $\mu$g m$^{-3}$ h$^{-1}$ in the winter, spring, summer, and autumn, respectively. The sharp decline hours of the net IPRs (around 11:00–18:00 pm) are mainly dominated by TRAN (sum of HTRA and VTRA) processes, with the mean rates of –1.4, –0.8, –0.7, and –0.3 $\mu$g m$^{-3}$ h$^{-1}$ in the winter, spring, summer, and autumn, respectively. However, in summer, TRAN processes constitute the dominant source during midnight (1:00–6:00 am), owing to higher $NO_3^-$ concentrations at the upper PBL contributing to the surface through the vertical mixing and development of the PBL (Huang et al., 2020c). In Fig. S9, VTRA processes act as the main positive contributor to $NO_3^-$ buildup production from 0:00 to 23:00 at layer 1 (surface layer), while AERO processes make the negative contribution to $NO_3^-$ within layers 1–8 (from the surface to 800 m). Above layer 10, AERO processes for $NO_3^-$ production are positive in the daytime, which is conducive to the accumulation of $NO_3^-$concentrations.

For $HNO_3$, a one-peak mode diurnal variation of the net IPRs rates is found, and peak times are at 20:00 pm in the winter and around 9:00–12:00 am in other seasons (Fig. 5). Meanwhile, CHEM (gas chemical processes) processes are the major



contributor to $HNO_3$ formation, with the peak rates being 0.6, 1.4, 2.3, and 0.7 ppb h$^{-1}$

in the winter, spring, summer, and autumn, respectively. In the spring, summer and

autumn, the peak times of $HNO_3$ formation are consistent with the first-peak times of

$NO_3^-$. The seasonal net IPRs rates reached a maximum of 0.3, 1.0, and 0.1 ppb h$^{-1}$,

respectively. CHEM and VTRA processes are the dominant contributors of $HNO_3^-$

production, especially during 7:00 to 13:00 (net IPRs rates > 0), with the seasonal peak

rates of 1.5, 2.7, and 0.8 ppb h$^{-1}$, respectively. AERO, DDEP, and HTRA processes are

the dominant contributors of the $HNO_3^-$ sharp decline (14:00–17:00 pm), with the

lowest net IPRs rates of –0.8, –0.7, and –0.3 μg m$^{-3}$ h$^{-1}$ in the spring, summer, and

autumn, respectively. DDEP processes are the dominant sink of $HNO_3$ in summer (–

0.64 ± 0.20 ppb h$^{-1}$). However, in the winter, the peak times of $HNO_3$ production are

opposite with the first-peak of $NO_3^-$, but consistent with the second-peak. HTRA make

a positive contribution to $HNO_3^-$, with peak rates of 0.18 ppb h$^{-1}$ at 20:00 pm. In Fig.

S12, the only-largest sink is the AERO process, consistent with efficient partitioning of

$HNO_3$ into particle phase $NO_3^-$ in cold seasons.

            Table 3 illustrates that within the PBL, in cold seasons (winter and autumn), about

60–78 % of $TNO_3$ is produced through OH+$NO_2$, 21–36 % through HET $N_2O_5$, and 2–

5 % through the "Others" pathways in the five representative YRD cities. Meanwhile,

71–83 % of $TNO_3$ is produced through OH+$NO_2$, 10–23 % through HET $N_2O_5$, and 4–

13 % through the "Others" pathways (mainly contributed by $NO_3$+Org and $N_2O_5$ $H_2O$)

in warm seasons (summer and spring). Table 4 shows the comparison of the contribution

of the major $TNO_3$ production pathways studies in China and other regions using

different methods. The results are in agreement with the contribution of $NO_3^-$ pathways

in previous modeling and observational studies. For example, Li et al. (2021b) modeled

that OH+$NO_2$ and HET $N_2O_5$ pathways dominate $NO_3^-$ production in the YRD region

in warm and cold seasons of 2016 by the CTM, accounting for 86–92 % and 8–14 % in

the surface layer, respectively. He et al. (2020) reported that the OH+$NO_2$ pathway



dominates $NO_3^-$ production in Shanghai on the surface layer using nitrogen isotopes analysis, accounting for 84–92 % and 55–77 % in the warm and cold seasons of 2016, respectively. Alexander et al. (2020) highlighted that the $OH+NO_2$ and HET $N_2O_5$ pathways contribute the same proportion (both 41 % in the four seasons) to $NO_3^-$

production in the global region using the CTM and oxygen isotopes analysis.

Fig. 6a shows the diurnal variations of $TNO_3$ formation reactions rates through three major pathways in Shanghai within the PBL. The average diurnal trends of $TNO_3$ production rates are consistent with the CHEM processes rates of $HNO_3$ production (Figs. 5–6). The chemical production of $HNO_3$ quickly transforms to particulate $NO_3^-$,

through AERO processes in the presence of abundant $NH_3$. In the winter, spring, summer, and autumn, the averaged $TNO_3$ production rates are $0.31 \pm 0.14$, $0.65 \pm 0.37$, $1.09 \pm 0.68$, and $0.28 \pm 0.22$ ppb h$^{-1}$, respectively (Table S6). Moreover, the seasonal peak rates of $TNO_3$ production are 0.6, 1.4, 2.3, and 0.7 ppb h$^{-1}$ around 11:00 am –13:00 pm, respectively. In accordance with the seasonal variation of $HNO_3$ net IPRs rates,

$TNO_3$ production rates are the fastest in summer.

In Shanghai, $TNO_3$ chemical production is dominated by the $OH+NO_2$ pathway on the daily timescale, accounting for 69.3–86.9 % of the total, while the HET $N_2O_5$ pathway is likewise a relatively important pathway (accounting for 11.1–28.4 %) in the four seasons (Fig. 6b). Notably, $TNO_3$ production rates are dominated by the $OH+NO_2$

pathway during the daytime (7:00 am–18:00 pm, accounting for 88.4–97.9 % of the total) in all seasons, while the HET $N_2O_5$ pathway becomes more important for the $TNO_3$ production during the nighttime (19:00 pm – 06:00 am, accounting for 42.5–61.6%). During winter, $TNO_3$ formation via the HET $N_2O_5$ pathway becomes dominant over the $OH+NO_2$ pathway, accounting for 62, 65, and 68% in Shanghai, Hangzhou and Nanjing at night, respectively. $O_3$ strongly coordinates $TNO_3$ production in YRD

via the HET $N_2O_5$ pathway during the nighttime. Similarly, He et al. (2018) observed that the HET $N_2O_5$ pathway was the major contributor to $NO_3^-$ production in the winter



of Beijing at the surface layer, using oxygen and nitrogen isotopes analysis, accounting for 56–97 % of the total during the nighttime. In another CTM study in the NCP, the HET $N_2O_5$ pathway was the dominant contributor to nocturnal-$NO_3^-$ production within the PBL in winter, with a contribution of 83 % at night (Liu et al., 2020a).

Fig. 7 displays the contributions of $TNO_3$ formation pathways from the local and transport (sum of indirect and direct transport) contributions. For the local contribution, the averaged $TNO_3$ production rates are $0.27 \pm 0.14$, $0.56 \pm 0.37$, $1.05 \pm 0.69$, and $0.26 \pm 0.21$ ppb $h^{-1}$ in the winter, spring, summer, and autumn, respectively (Table S8). During the daytime, the $OH+NO_2$ pathway contributes almost all $TNO_3$ production rates from the local contribution, accounting for about 89–98 % of the total, with mean rates of $0.33 \pm 0.17$, $0.83 \pm 0.34$, $1.55 \pm 0.59$, and $0.40 \pm 0.22$ ppb $h^{-1}$ in the winter, spring, summer, and autumn, respectively. The results suggest that the locally-emitted $NO_2$ reacts with locally-formed OH dominated $TNO_3$ production rates during the day in the urban YRD region. For the transport contribution, the averaged $TNO_3$ production rates are $0.04 \pm 0.01$, $0.08 \pm 0.02$, $0.03 \pm 0.02$, and $0.02 \pm 0.01$ ppb $h^{-1}$ in the winter, spring, summer, and autumn, respectively (Table S9). The HET $N_2O_5$ pathway is noted as the dominant pathway for $TNO_3$ production of the transport contribution, accounting for around 72–86 % during the nighttime. Fig. 7b compares the $TNO_3$ production pathways rates between indirect and direct transport contributions. The regional production is mainly contributed by indirect transport, especially in the winter and summer. The results suggest that the transported $O_3$ from outside YRD region can react with the locally-emitted $NO_2$, supporting $HNO_3$ formation production via the HET $N_2O_5$ chemistry pathway at night.

## 4. Conclusions

This study investigates the contributions of regional transport and major chemical pathways to the of $NO_3^-$ and $HNO_3$ formation in YRD in different seasons using the

WRF-CMAQ model. The modeled results show that local emissions dominate YRD-

regional $NO_3^-$ concentrations (50–62%), while regional transport contributes 38–50%

to $NO_3^-$ (indirect transport contributes 24–37%). Except for winter, $HNO_3$ was

dominated by the contributions of local emissions (61–75%) and indirect transport

contributed negatively –24 to –41%. In Shanghai, the IPRs analysis reveals that AERO

processes were the predominant contributors in $NO_3^-$ formation within the PBL. TRAN

processes were the largest sinks in $NO_3^-$ formation in the winter, spring and autumn,

while the positive contributors at night in summer. For $HNO_3$, CHEM processes were

the only positive contributor during the day. The $OH+NO_2$ pathway is the predominant

contributor (60–83%) among all chemical pathways, while the HET $N_2O_5$ pathway is

also important (10–36%) in YRD. The $TNO_3$ production is dominated by the $OH+NO_2$

pathway during the day (98%) in summer, while the HET $N_2O_5$ pathway dominates

during the night (61%) in winter. The $TNO_3$ production rates from the local and

transport contributions were further elucidated. The $OH+NO_2$ pathway from the local

contribution strongly dominates the $TNO_3$ production during the day (89–98%). At

night, the HET $N_2O_5$ pathway mainly dominate by indirect transport (via reaction with

transported $O_3$ at night).

### Code and data availability

Hourly concentrations of $O_3$, $PM_{2.5}$, $NO_2$, $SO_2$, and CO used in this study are freely

available through the website of http://106.37.208.233:20035/ (last accessed on April

30, 2022). Observation data of meteorological parameters used in this study are

available from http://data.cma.cn/en (last accessed on November 30, 2021). The CMAQ

outputs are currently available upon request, all python codes used to create any of the

figures are available upon request.



## Author contributions

JS, MQ and JH designed research. JS, MQ, XX, WF, YQ, LS, and LL contributed to model development, simulations, and data processing. JL, IS, LJ, LH, XY contributed to result discussion. JS prepared the manuscript and all coauthors helped improve the manuscript.

## Competing interests

The authors declare that they have no conflict of interest.

## Acknowledgements

This work was supported by the National Natural Science Foundation of China (42007187, 92044302, 42021004), and the Postgraduate Research & Practice Innovation Program of Jiangsu Province (KYCX21_0954).

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



## Tables and Figures

**Table 1.** Model performance for meteorological parameters for January, April, July, October and the annual average of 2017 in the entire YRD region. The values that do not meet the criteria are denoted in bold.

| Parameters | Statistic(benchmarks) | January | April | July | October | Annual |
|---|---|---|---|---|---|---|
| T2(°C) | MB ($\leq \pm 0.5$) | **1.56** | **1.04** | **0.67** | **1.98** | **1.31** |
| | RMSE | 1.99 | 1.76 | 1.57 | 2.24 | 1.89 |
| | IOA ($\geq 0.8$) | 0.89 | 0.93 | 0.85 | 0.89 | 0.89 |
| | R | 0.94 | 0.93 | 0.85 | 0.95 | 0.92 |
| RH(%) | MB | -6.96 | -10.70 | -9.06 | -5.98 | -8.17 |
| | RMSE | 9.73 | 13.14 | 10.91 | 8.02 | 10.45 |
| | IOA | 0.88 | 0.83 | 0.72 | 0.82 | 0.81 |
| | R | 0.90 | 0.91 | 0.88 | 0.87 | 0.89 |
| WD(°) | MB ($\leq \pm 10$) | **-12.78** | -0.92 | **12.26** | **-24.42** | -6.46 |
| | RMSE | 37.68 | 36.04 | 26.61 | 55.85 | 39.05 |
| | IOA | 0.88 | 0.89 | 0.88 | 0.76 | 0.85 |
| | R | 0.85 | 0.82 | 0.85 | 0.70 | 0.81 |
| WS(m/s) | MB ($\leq \pm 0.5$) | **0.61** | **0.76** | **1.03** | **0.69** | **0.77** |
| | RMSE ($\leq 2.0$) | 0.82 | 1.06 | 1.31 | 0.96 | 1.04 |
| | IOA ($\geq 0.6$) | 0.84 | 0.71 | 0.65 | 0.82 | 0.76 |
| | R | 0.89 | 0.75 | 0.75 | 0.88 | 0.82 |

Notes: The following equations of MB, RMSE and IOA are defined in Table S4. The benchmarks are suggested by Emery and Tai (2001).





**Table 2.** Model performance of major pollutants for the full year of 2017 in five representative YRD cities [a].

| [b] Pollutants | Shanghai | | | | Nanjing | | | | Hefei | | | | Hangzhou | | | | Changzhou | | | |
| --- | --- | --- | --- | --- | --- | --- | --- | --- | --- | --- | --- | --- | --- | --- | --- | --- | --- | --- | --- | --- |
| | [c] NMB | NME | MB | R | NMB | NME | MB | R | NMB | NME | MB | R | NMB | NME | MB | R | NMB | NME | MB | R |
| MDA8 $O_3$ | -0.01 | 0.20 | -1.07 | 0.88 | **0.17** | **0.28** | 18.59 | 0.76 | **0.17** | 0.24 | 17.23 | 0.81 | **0.25** | **0.31** | 26.60 | 0.80 | **0.19** | **0.26** | 19.85 | 0.84 |
| $NO_2$ | -0.05 | 0.23 | -1.89 | 0.71 | -0.07 | 0.26 | -3.20 | 0.50 | -0.11 | 0.26 | -5.21 | 0.67 | -0.25 | 0.34 | -10.37 | 0.51 | -0.07 | 0.24 | -2.67 | 0.56 |
| $SO_2$ | -0.38 | 0.43 | -4.61 | 0.66 | 0.12 | 0.45 | 1.83 | 0.32 | 0.01 | 0.36 | 0.18 | 0.75 | -0.28 | 0.40 | -3.15 | 0.46 | 0.09 | 0.34 | 1.54 | 0.48 |
| CO | -0.38 | 0.40 | -0.29 | 0.67 | -0.17 | 0.33 | -0.17 | 0.45 | -0.22 | 0.26 | -0.19 | 0.76 | -0.30 | 0.34 | -0.25 | 0.55 | 0.06 | 0.25 | 0.05 | 0.64 |
| $PM_{2.5}$ | -0.08 | 0.30 | -2.80 | 0.73 | 0.28 | 0.44 | 10.29 | 0.75 | **0.41** | **0.51** | 21.42 | 0.76 | 0.05 | 0.31 | 1.88 | 0.69 | 0.25 | 0.37 | 10.59 | 0.78 |
| $NO_3^-$ | -0.49 | 0.63 | -3.25 | 0.70 | | | | | 0.07 | 0.65 | 0.32 | 0.59 | 0.11 | 0.79 | 0.64 | 0.43 | -0.36 | 0.58 | -3.34 | 0.56 |

Notes: [a] The year of 2017 includes the four typical months (January, April, July, and October). [b] MDA8 $O_3$, $NO_2$, $SO_2$ and $PM_{2.5}$ units (μg/m³), CO units (mg/m³).

[c] The equations of NMB, NME, MB and R are found in Table S4. The values that do not meet the criteria are highlighted in bold.

The recommended benchmarks for MDA8 $O_3$, 24-h $PM_{2.5}$ and $NO_3^-$ are suggested by Emery et al. (2017) and Huang et al. (2021).





**Table 3.** Model performance for production rates (ppb/h) and daily contributions in percentage of the major production pathways (%) for the four seasons of 2017 in five representative YRD cities.

| Selected cities | Seasons | $TNO_3$ | $OH\ NO_2$ | $HET\ N_2O_5$ | $OH\ NO_2$ (%) | $HET\ N_2O_5$ (%) | Others (%) |
|---|---|---|---|---|---|---|---|
| Shanghai | Winter | 0.31 ± 0.13 | 0.21 ± 0.18 | 0.09 ± 0.06 | 69.3% | 28.4% | 2.2% |
| | Spring | 0.65 ± 0.35 | 0.52 ± 0.43 | 0.10 ± 0.09 | 81.8% | 15.3% | 2.9% |
| | Summer | 1.09 ± 0.68 | 0.90 ± 0.80 | 0.13 ± 0.15 | 82.9% | 12.2% | 4.9% |
| | Autumn | 0.28 ± 0.22 | 0.24 ± 0.24 | 0.03 ± 0.03 | 86.9% | 11.1% | 2.0% |
| Nanjing | Winter | 0.38 ± 0.13 | 0.23 ± 0.20 | 0.14 ± 0.11 | 59.2% | 36.1% | 4.7% |
| | Spring | 0.65 ± 0.29 | 0.48 ± 0.40 | 0.14 ± 0.12 | 73.1% | 21.4% | 5.4% |
| | Summer | 0.83 ± 0.41 | 0.62 ± 0.55 | 0.15 ± 0.17 | 74.7% | 17.9% | 7.4% |
| | Autumn | 0.50 ± 0.25 | 0.35 ± 0.32 | 0.13 ± 0.11 | 69.7% | 25.4% | 4.9% |
| Hefei | Winter | 0.38 ± 0.13 | 0.26 ± 0.18 | 0.10 ± 0.07 | 66.9% | 27.1% | 6.0% |
| | Spring | 0.63 ± 0.24 | 0.49 ± 0.30 | 0.10 ± 0.09 | 78.5% | 16.5% | 5.0% |
| | Summer | 0.66 ± 0.26 | 0.54 ± 0.30 | 0.07 ± 0.08 | 81.7% | 10.4% | 7.9% |
| | Autumn | 0.48 ± 0.18 | 0.35 ± 0.24 | 0.11 ± 0.08 | 72.5% | 21.8% | 5.7% |
| Changzhou | Winter | 0.41 ± 0.15 | 0.29 ± 0.20 | 0.11 ± 0.08 | 68.9% | 26.8% | 4.3% |
| | Spring | 0.64 ± 0.25 | 0.48 ± 0.31 | 0.13 ± 0.12 | 74.9% | 20.9% | 4.2% |
| | Summer | 0.70 ± 0.27 | 0.55 ± 0.31 | 0.10 ± 0.13 | 78.7% | 14.3% | 7.0% |
| | Autumn | 0.46 ± 0.19 | 0.36 ± 0.24 | 0.08 ± 0.07 | 77.6% | 18.3% | 4.1% |
| Hangzhou | Winter | 0.43 ± 0.15 | 0.26 ± 0.21 | 0.15 ± 0.12 | 59.7% | 35.5% | 4.8% |
| | Spring | 0.57 ± 0.24 | 0.40 ± 0.33 | 0.13 ± 0.12 | 70.5% | 23.3% | 6.2% |
| | Summer | 0.47 ± 0.23 | 0.36 ± 0.29 | 0.05 ± 0.05 | 76.4% | 10.7% | 12.9% |
| | Autumn | 0.46 ± 0.26 | 0.34 ± 0.32 | 0.10 ± 0.09 | 73.8% | 21.3% | 4.9% |





**Table 4.** Comparison of contributions of major nitrate formation pathways in China and others regions [a].

| References | Methods [b] | Study seasons | Year | Study regions | $NO_3^-$ formation pathways [c] | Time metric | Contribution (%) |
|---|---|---|---|---|---|---|---|
| (Li et al., 2021b) | WRF-Chem | Warm (Aug –Sep) Cold (Nov-Dec) | 2016 | NCP, YRD | OH+NO2 (layer 1) HET N2O5 (layer 1) | season-mean | 60-92% 8-40% |
| (Qu et al., 2021) | WRF-CMAQ PA | Transition season (Oct-Dec) | 2015 | PRD | OH+NO2 (layers 1-4) HET N2O5 (layers 1-4) | day-mean night- mean | 92-96% 64-72% |
| (Chuang et al., 2022) | WRF-CMAQ PA | Transition season (Mar - Apr) | 2017 | Taiwan | OH+NO2 HET N2O5 | day-mean night- mean | > 90% 30-90% |
| (Wu et al., 2021) | WRF-Chem; Nitrogen Isotopes | Cold (Dec-Jan) | 2017 | Xi'an | HET N2O5 (surface) | season-mean | 13-35 % |
| (Chan et al., 2021) | GEOS-Chem; Isotope tracing | Cold | 2014-15 | NCP | OH+NO2&HET N2O5 (surface) | season-mean | 34 % & 45 % |
| (Fu et al., 2020) | WRF-CMAQ PA | Cold (Dec) | 2017 | NCP | OH+NO2 (HET N2O5) 10 layers | season-mean | 43% (44%) |
| (Liu et al., 2020a) | WRF-Chem | Cold (Dec) | 2016 | NCP | HET N2O5 (surface) HET N2O5 (PBL) | haze-mean night (day) | 52 % 83% (10%) |
| (Zhang et al., 2021) | Nitrogen &Oxygen Isotopes | Cold (Nov-Jan) | 2017-18 | Nanchang | HET N2O5 (surface) | season-mean | 60% |
| (Fan et al., 2021) | Nitrogen &Oxygen Isotopes | Warm and Cold | 2016-17 | Beijing | OH+NO2&HET N2O5 (260 m) | Clean days | 20% (80%) |
| (Luo et al., 2020a) | Nitrogen &Oxygen Isotopes | Spring(Mar-May) | 2013 | Beijing | OH+NO2 (surface) OH+NO2 (surface) | Clean days Polluted days | 24-50% 11-47% |
| (Luo et al., 2020b) | Nitrogen &Oxygen Isotopes | Four seasons | 2018 | Nanchang | OH+NO2 (HET N2O5) | season-mean | 12-59% (67-89%) |
| (Fan et al., 2020) | Nitrogen &Oxygen Isotopes | Cold (Nov-Dec) | 2018 | Beijing | HET N2O5 | haze period | 64% |
| (He et al., 2020) | Nitrogen &Oxygen Isotopes | Warm and Cold season | 2016 | Shanghai | OH+NO2 (warm) OH+NO2 (cold) | season-mean | 84-92% 48-74% |
| (Wang et al., 2019) | Nitrogen &Oxygen Isotopes | Warm and Cold season | 2014 | Beijing | OH+NO2 HET N2O5 | annual-mean annual-mean | 32 ± 10% 68 ± 23% |
| (He et al., 2018) | Nitrogen isotopic | Cold (Oct-Jan) | 2014 | Beijing | HET N2O5 | night-haze | 56-97 % |
| (Chen et al., 2020) | Field determination; Box model | Cold (Nov-Dec) | 2016-17 | Beijing | OH+NO2& HET N2O5 (240 m) | haze period | 74-76% & 34% |
| (Zang et al., 2022) | Field observations; Box model | Cold (Dec-Feb) | 2018-19 | Shanghai (urban | OH+NO2&HET N2O5 (surface) | haze period | 69% & 29% |




| | | | | & suburban areas) | OH+$NO_2$&HET $N_2O_5$ (surface) | haze period | 63% & 35% |
|---|---|---|---|---|---|---|---|
| (Womack et al., 2019) | Box model | Cold (Dec) | 2016-17 | Salt Lake Valley | HET $N_2O_5$ (RL) | season-mean | 43% |
| (Vrekoussis et al., 2004) | Field determination, Box model | Summer(Jul-Aug) | 2001 | South-East Europe | HET $N_2O_5$ (surface) | season-mean | 21% |
| (Kim et al., 2014) | WRF-CMAQ PA | Cold (Dec) | 2009 | The Great Lakes | OH+$NO_2$&HET $N_2O_5$ (surface) | season-mean | 28% & 57% |
| (Shah et al., 2018) | GEOS-Chem | Cold (Feb-Mar) | 2015 | Eastern US | OH+$NO_2$&HET $N_2O_5$ (surface) | season-mean | 36% & 62% |
| (Alexander et al., 2020) | GEOS-Chem; Oxygen Isotopes | Four seasons | 2000-15 | Global | OH+$NO_2$ (below 1 km) | annual-mean | 41-42% |
| | | | | | HET $N_2O_5$ (below 1 km) | | 28-41% |

Notes: [a] The above studies are conducted in the major regions and megalopolises of China (the North China Plain (NCP), Yangtze River Delta (YRD), Pearl River

Delta (PRD)), the United States, and the Global region. The comparison serves to quantify the relative contribution of two main nitrate formation pathways in different

seasons. [b] Methods include the 3-D CTMs, nitrogen and oxygen isotopes analysis, field determination, and box model. [c] Surface represents the surface layer.



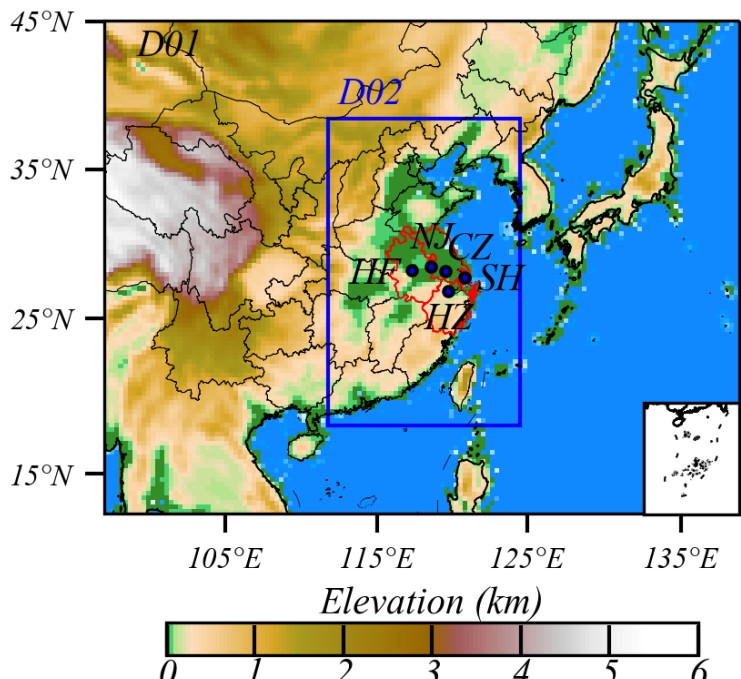

779

**Fig 1.** Entire YRD region as the target region (marked as red) in two nested simulation
domains (36 and 12 km resolutions), and location of five representative YRD cities
used in modeling evaluations in the d02 modeling domain.

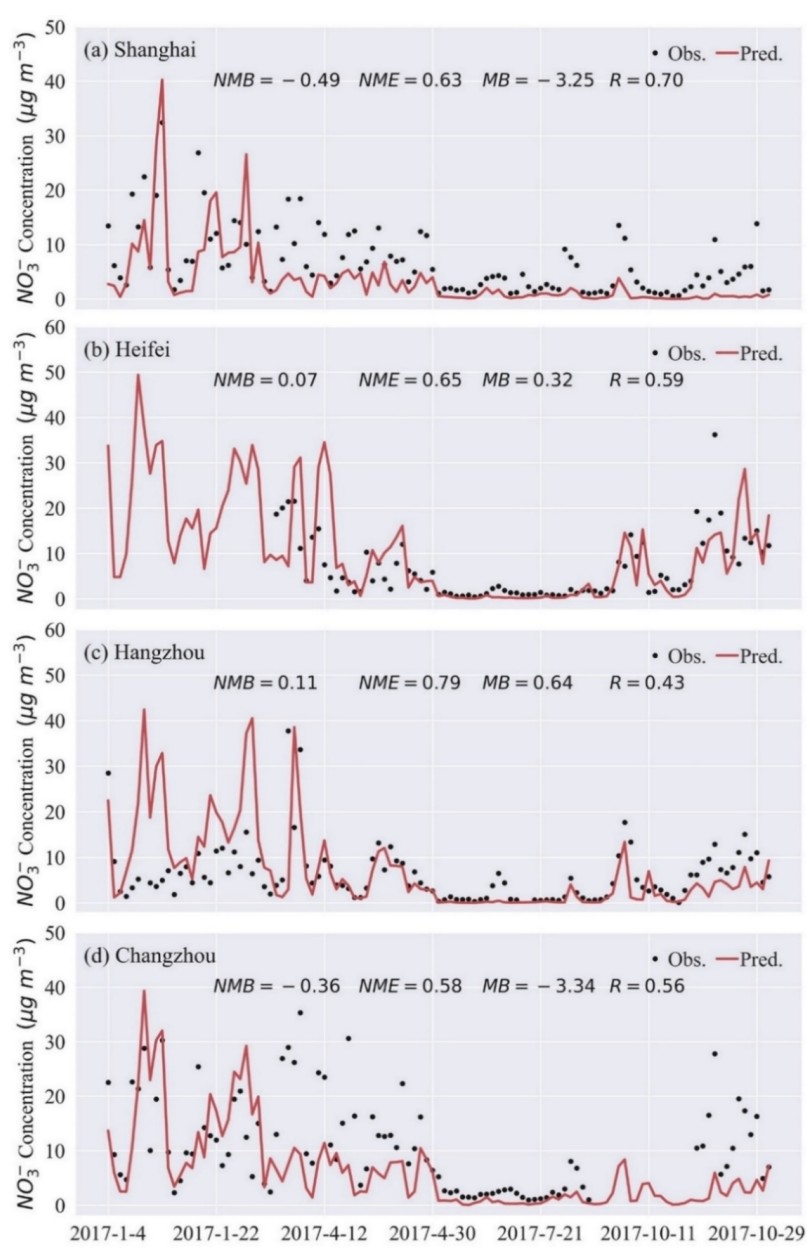

**Fig 2.** Time series of predicted (red) and observed (black) daily NO₃⁻ concentrations in four atmospheric environment supersites (a–d) in January, April, July, and October 2017.



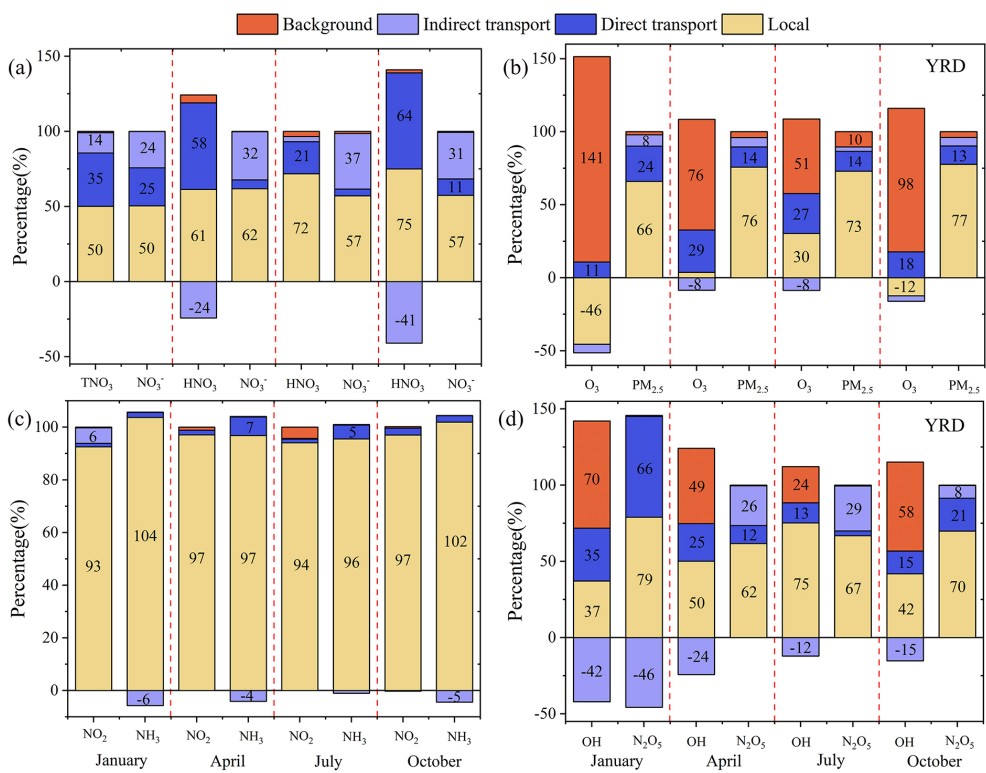

**Fig 3.** (a–d) Contributions of Background, Local, Indirect, and Direct transport to nitrate-related species in four months of 2017 for the entire YRD region.

Notes: Nitrate-related species represent $NO_3^-$, $HNO_3$, $PM_{2.5}$, $O_3$, $NO_2$, $NH_3$, OH, and $N_2O_5$. The contributions of $HNO_3$ in January 2017 are shown in Fig. S6.

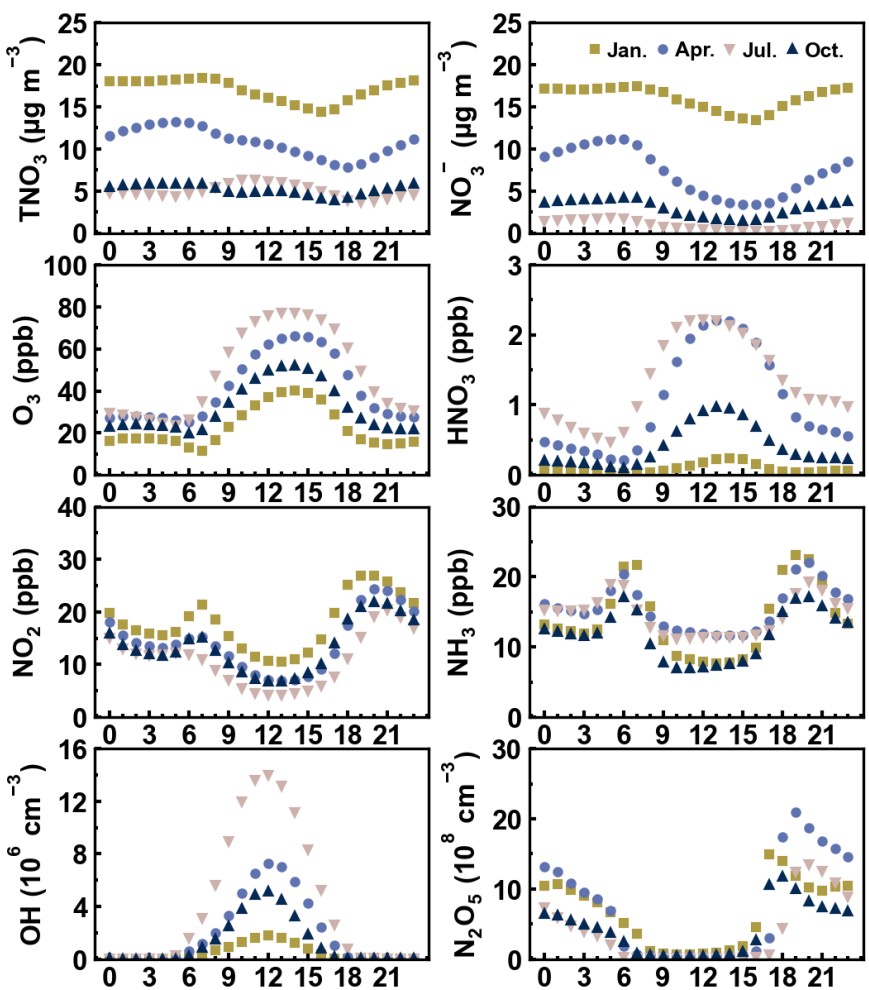

**Fig 4.** Monthly diurnal variations of three nitrate-phases ($NO_3^-$, $HNO_3$, and $TNO_3$), major nitrate-precursors ($O_3$, $NO_2$, and $NH_3$) and major atmospheric oxidants (OH and $N_2O_5$) for the entire YRD region under the base scenario. The X axis marks each hour of the day (Beijing time).

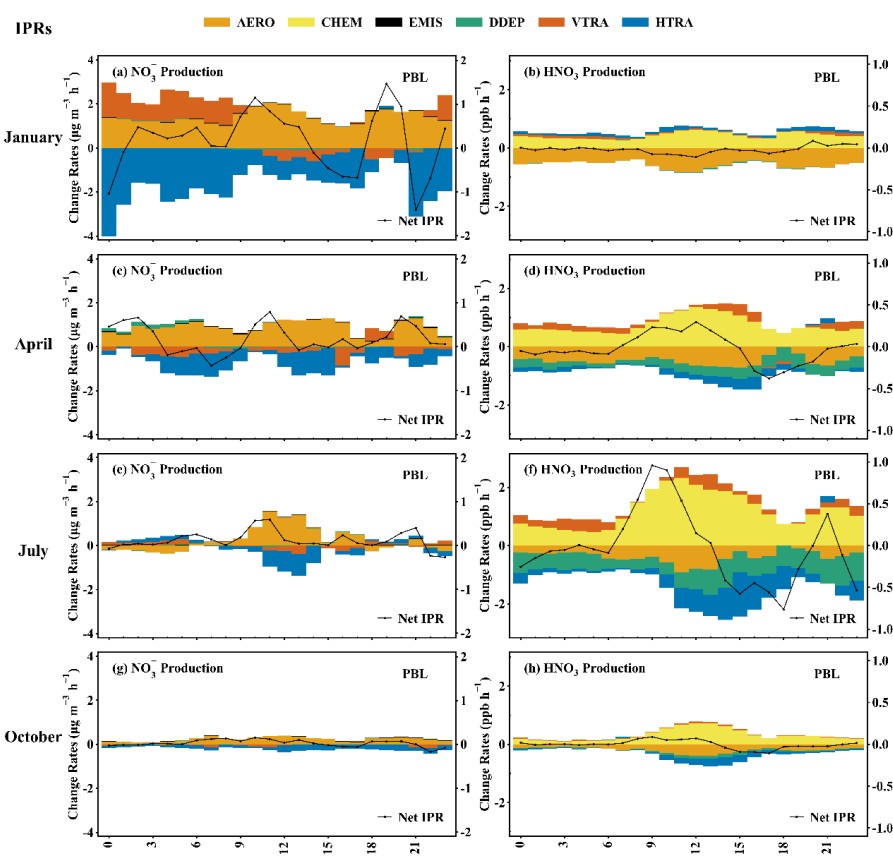

**Fig 5.** Diel variations in physical and chemical processes rates of $NO_3^-$ and $HNO_3$ production (a–h) within the PBL in Shanghai. Black line represents the net IPR value for each hour of the day; its value scale is on the right Y axis.



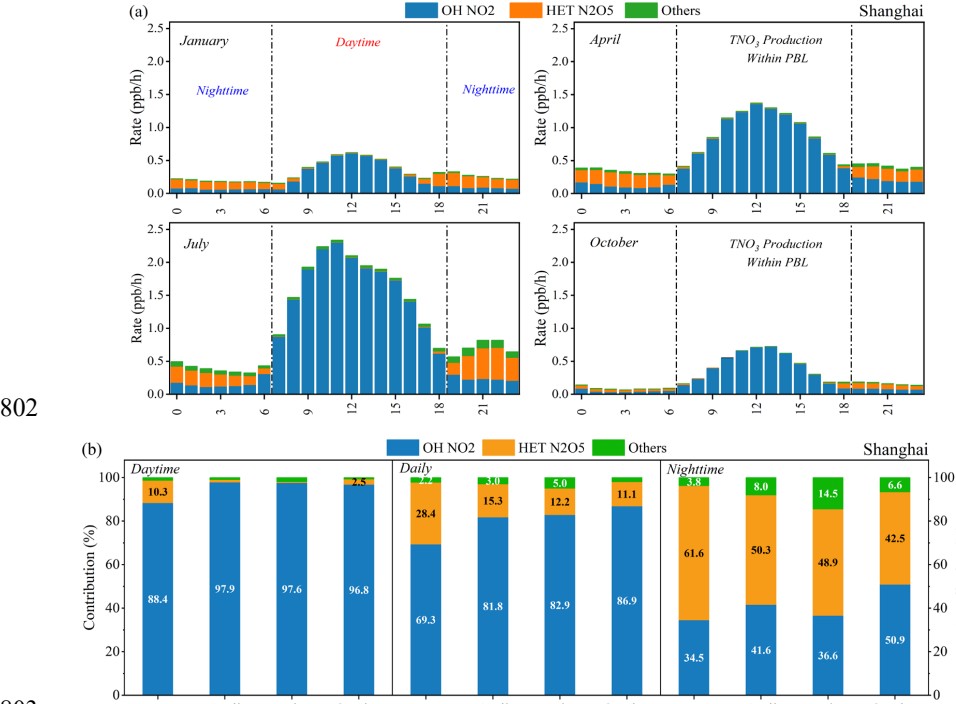

**Fig 6.** (a) Mean diurnal variations of $TNO_3$ production rates in different pathways in 2017 in Shanghai. (b) Average potential contribution of $OH + NO_2$, HET $N_2O_5$ and Others pathways to $TNO_3$ chemical production in Shanghai within the PBL under base case simulation.

Notes: Daytime (7:00–18:00), Nighttime (19:00–6:00). $OH + NO_2$ and HET $N_2O_5$ pathways are noted as "OH NO2" and "HET N2O5" in Figs.6 and 7.

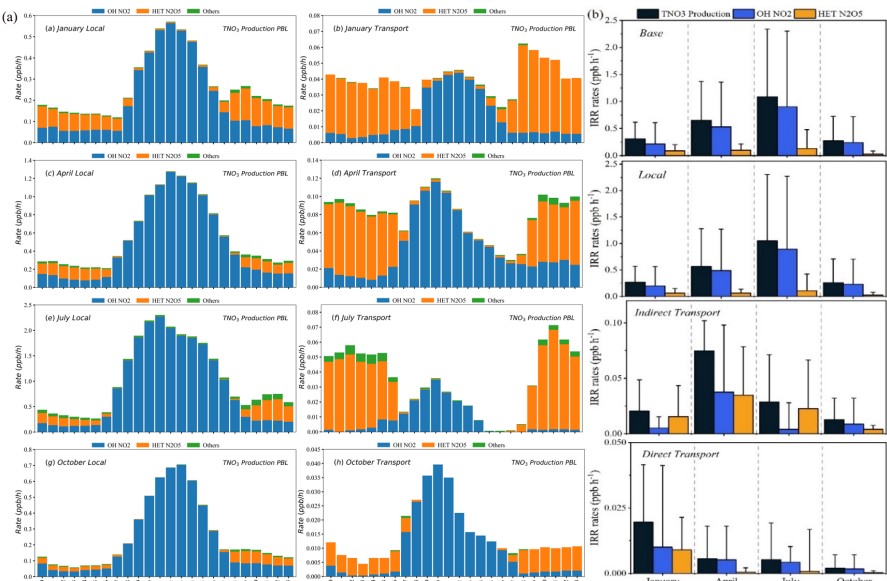

814

**Fig 7.** (a) Mean diurnal variations of TNO$_3$ production rates in major pathways from
the local and transport (sum of indirect and direct transport) contributions. (b) Total
IRRs rates of TNO$_3^-$ production rates in the base case and from the local and transport
contributions within the PBL. The error bar indicates one standard deviation.