# Peer review of "Seasonal modeling analysis of nitrate formation pathways in Yangtze River Delta region, China"

_Atmospheric Chemistry and Physics, 2022_

## Author Response (AR1)

Dear Editor and Reviewers,

Thank you for the comments to help improve the quality of the paper. We have revised the manuscript to address your comments and a detailed response to each comment is provided in this file. The comments are in regular font and the responses are in red.

Seasonal modeling analysis of nitrate formation pathways in Yangtze River Delta region, China
Manuscript #: acp-2022-426

**RC1, Reviewer #1:**

Nitrate has been the dominant chemical component of PM2.5 in China during winter haze days in recent years. This manuscript used the CMAQ air quality model to investigate the impact of local emission and regional transport on nitrate formation as well as its major formation pathways in the Yangtze River Delta (YRD) region during the four seasons of 2017. Overall, the results are interesting and meaningful for future emission control strategy design. The manuscript is well written and easy to follow. I recommend accepting this manuscript after some minor revisions.

Comments:
1) Lines 277-282: From Figure 2, the model performance in Hefei is better than the other three sites. Why do the authors only focus on the results in Shanghai, Hangzhou, and Changzhou?

Responses: The statistics in Figure 2b show that Heifei has the best performance on nitrate modeling throughout the year. However, the measurement of nitrate in Hefei is not available in January, when nitrate shows the highest concentration and is of most concern. As the simulation of nitrate has been evaluated in winter in Shanghai, Hangzhou, and Changzhou (see Figure R1), we performed the analysis for the three cities. We clarified in the manuscript in Lines 279-285:

"The daily concentrations of $NO_3^-$ are efficiently predicted in four supersites, all within the benchmark (NMB $\leq \pm 0.60$, NME $\leq 0.75$, and R > 0.6). But in Hefei (Fig. 2b), the wintertime $NO_3^-$ measurement data is not available, when $NO_3^-$ shows the highest concentrations and is of most concern. Good agreement between predicted and observed values is demonstrated on daily timescales, especially in Shanghai (NMB = –0.49, R = 0.70), Hangzhou (NMB = 0.11, MB = 0.64) and Changzhou (NMB = 0.36, R = 0.56).".

[Figure]

Figure R1. Hourly time series of prediction (red) and observation (black) particulate $NO_3^-$ concentrations in four urban monitoring sites in winter of 2017.

2) Lines 283-284: Underestimation of $NO_3^-$ can also be found in spring, and the bias may even be larger than that in summer and autumn. It's better to provide the model performance in each season.

Responses: Thanks for the suggestion. We have evaluated nitrate simulation in each season and added Figure S2 in the Supplementary material (SI). We mentioned this in Lines 287-289: "Fig. S2 shows the hourly predicted and observed $NO_3^-$ concentrations in each season. $NO_3^-$ concentrations are generally underestimated during the summer and autumn.".

3) Lines 313-314: Local emission only contributes negatively in winter and autumn, shouldn't be "in the four seasons".

Responses: We have corrected the manuscript, see Lines 320-321: "For $O_3$, the local emissions have negative contribution in winter (−46%) and autumn (−12%).".

4) Lines 315-317: The indirect transport doesn't seem to contribute as large as −7% in autumn according to Figure 3b. Please check the numbers.

Responses: We have carefully checked the numbers in Figure 3b, where −7% represents the mean contributions of the four seasons. The sentences have been rephrased in the manuscript, see Lines 321-323: "The negative contributions of the indirect transport are –6, –8, –8, and –4 % in winter, spring, summer, and autumn, respectively.".

5) Line 317: Avoid such expressions as "−12%–−42%".
Responses: Corrected. We have revised the manuscript, as following in Lines 323-324: "In Fig. 3d, the indirect transport contributes from –42% to –12% of OH concentrations in the four seasons. ".

6) Line 70: Please check the format of this paragraph.
Responses: Thanks for the suggestion. We have checked the format of this paragraph. The manuscript has been revised as following in Lines 69-73:
"Owing to the stringent emission control strategies since 2013, primary $PM_{2.5}$, the major precursors (i.e., sulfur dioxide ($SO_2$) and nitrogen oxides ($NO_x$ = nitric oxide (NO) + nitrogen dioxide ($NO_2$)) emissions have decreased substantially in China, which led to significant decreases in total $PM_{2.5}$ and sulfate ($SO_4^{2-}$) mass concentrations.".

7) Lines 373, 376, and 381: "$HNO_3^-$" should be "$HNO_3$".
Responses: Corrected. The manuscript has been carefully revised the manuscript in Lines 379, 382, and 387.

8) Line 415: Providing TNO3 production rates from different pathways at different model layers would be more helpful.
Responses: Thanks for the advice. We added a figure in the SI (Fig.S8) with a few lines of discussion regarding $TNO_3$ production rates at different model layers in the manuscript (see Lines 432 to 440):
"As shown in the revised Figure S8, the seasonal $TNO_3$ production rates (ppb/h) and contributions (%) of the major pathways have been compared between vertical layers and PBL. The $OH+NO_2$ pathway dominated $TNO_3$ production at all layers, accounting for more than 58%, 78%, 80%, and 83% in winter, spring, summer, and autumn, respectively. The $OH+NO_2$ pathway rate decreases with altitude at vertically layers, where its contribution decreases from 87% to 58%, from 91% to 78%, from 93% to 80%, and from 95% to 83% in the four seasons, respectively. The HET $N_2O_5$ pathway becomes more important for the $TNO_3$ production within layers 4~8 (250 to 580 m) in winter, accounting for 37% (Fig. S8b).".

[Figure]

Fig. S8. Comparison of TNO₃ production rates (ppb/h) and contributions (%) of three major pathways between vertical layers (Layer 1, 2, 4, 6, 8, 10 and 12, corresponding to the height ranging from 32 to 80 m, 80 to 160 m, 250 to 320 m, 420 to 500 m, 580 to 760 m, 1000 to 1300 m, and 2000 to 2800 m, respectively) and PBL.

9) Figure 7: The text in the figure is too small. Please make it larger.

Responses: Corrected. Figure 7 (a-b) has been disassembled into Figures 7 and 8 in the revised manuscript, and the manuscript has been revised accordingly in Lines 441-459.

10) Lines 440-445: The indirect transport of nitrate can also be formed from the transported HNO3 from outside YRD region reacting with the locally-emitted NH3. As can be seen from Figure 3, direct transport contributes considerably to HNO3. The authors should clarify this in the revision.

Responses: In Lines 195-201, the indirect transport contribution ($F_{indirect}$) represents that $NO_3^-$ generated from chemical reactions between outside-transported and local-emitted precursors, as calculated in Eq. (4). Hence, $F_{indirect}$ is mainly formed via two chemical pathways (the $OH+NO_2$ and HET $N_2O_5$ pathway), not including formed via by transported $HNO_3$ and local-emitted $NH_3$. Figures 7 and 8 suggest that the transported $O_3$ from outside YRD region react with the locally-emitted $NO_2$,

supporting $TNO_3$ production via the HET $N_2O_5$ chemical pathway at nighttime. We have added definition about $F_{indirect}$ in the manuscript (see Lines 195 to 197): "indirect transport ($NO_3^-$ contributed by transported and local-emitted precursors via the OH+$NO_2$ and HET $N_2O_5$ chemical pathway)".

11) Line 772: Table 3 is not for "model performance".
Responses: Corrected. Table 3 has been modified in the manuscript in Line 821.

12) Line 106: Actually, atmospheric oxidant doesn't include N2O5.
Responses: Corrected.

**RC2, Reviewer #2:**

The study provides a comprehensive overview of the pathways of nitrate formation in the Yangtze River Delta region. The study is of great interest and adds to the knowledge of the chemistry involved in the $NO_3^-$ formation and factors affecting these pathways for an urbanized and heavily populated region. The study is of importance as it provides a seasonal analysis of the $NO_3^-$ formation process which can be highly useful for regulators for planning and mitigation strategies. The paper is well written easy to follow. However, following are a few comments and suggestions which I think will help in improving the overall clarity of the paper.

Major comments:

1) The authors should highlight the reason for using SAES emission inventory over the YRD region instead of the MEIC or REAS emission inventory. The authors should also mention the resolution at which the SAES inventory provides emissions.

Responses: Thanks for the comment. In this manuscript, the anthropogenic emissions for the 2017 YRD region were released by the Shanghai Academy of Environmental Sciences (SAES) (An et al., 2021), a high-resolution (4 km × 4 km) air pollutant emission inventory for the entire YRD region. An et al. (2021) revealed that the SAES emission inventory was updated by using emission factors, and $PM_{2.5}$ and NMVOC speciation profiles based on local measure in the YRD region. An et al. (2021) also compared with the MEIC inventory , found that the gases precursors ($SO_2$ and $NO_x$) emissions estimated in the SAES emission inventory were lower and more realistic, and particulate matter emissions were higher due to the consideration of dust sources. Moreover, the SAES emission inventory has been thoroughly evaluated in our previous works (Liu et al., 2020;Qin et al., 2021;Li et al., 2021). In those previous studies, the statistical results of predictied $PM_{2.5}$, $NO_2$, and $O_3$ concentrations show a better model performance in the YRD region.

Above information about the SAES emission inventory have been added in the revised manuscript (see Lines 166-168): "The anthropogenic emissions for the 2017 YRD region were established by the Shanghai Academy of Environmental Sciences (SAES), a high-resolution (4 km × 4 km) anthropogenic emission inventory across the entire YRD region (An et al., 2021).".

2) Line 271-273: The authors may provide a reason for the better performance of the model in predicting the concentrations of $PM_{2.5}$, $O_3$ and $NO_2$ when compared to past studies.

Responses: Table 2 and Figure S1 indicated that the seasonal results of $PM_{2.5}$, $O_3$ and $NO_2$ show a better model performance than our previous works. We think the most likely reason is that we used the local anthropogenic emission inventory (SAES, with high-resolution and based on local measure data) to simulate the YRD region. As stated in the manuscript, our meteorological peroformance is comparable to previous

studies. But in this study, we used the local emission inputs. As explained in our first response, modeling results using the finer resolution YRD local emissions showed a better model performance in the YRD region when compared to using the MEIC emission inventory.

3) The authors may highlight key mitigation strategies for $NO_3^-$, based on the pathways identified as major contributors to $NO_3^-$ in the YRD region.

Responses: Thanks for the suggestion. Our findings illustrate that local emissions dominate $NO_3^-$ formation in the YRD (50–62%), more specifically, locally-emitted NOx reacting with OH and partitioning into particles with $NH_3$ (mostly from local sources, more than 93.0%), indicating that the coordinated control of precursors (i.e., NOx and $NH_3$) and reduction of the oxidative capacity of the atmosphere is crucial for $NO_3^-$ reduction.

Furthermore, regional transport contributes 38–50% to $NO_3^-$ formation in the YRD region. Indirect transport contributes 24–37% through transported $O_3$ reacting with local NOx at night, indicating that the simultaneous controlling of $O_3$ and $NO_3^-$ in the larger scale region is also important for $NO_3^-$ reduction in the YRD.

Above discussion have been added in the revised manuscript in lines 460-468.

4) The authors should add a section highlighting the limitations of the current study.

Responses: Limitations of the current study have been added in the revised manuscript in Lines 469-484.

This manuscript investigated the seasonal variations in the $NO_3^-$ formation mechanisms, including local emission and regional transport contributions, as well as dominant processes and major chemical pathways in the YRD region. However, there are still some limitations in this manuscript, such as the insufficient heterogeneous chemistry on the dust particles' surface and uncertainties in precursors emissions in the CMAQ model affect the model performance of $NO_3^-$ during the spring and autumn (Xie et al., 2022). Furthermore, the Integrated Reaction Rate (IRR) analysis was employed to quantify the rates of $TNO_3$ (sum of $NO_3^-$ and $HNO_3$) chemical reactions pathways, which potentially lead to differences in chemical pathways rates and contributions between $NO_3^-$ and $TNO_3$. Figure 6(b) illustrates that $TNO_3$ chemical production is dominated by the $OH+NO_2$ pathway on the daily timescale, accounting for 69.3–86.9 % in Shanghai. Notably, due to the higher temperature during the daytime, the potential production for $NO_3^-$ is not as high as that of the nocturnal chemical pathway (mainly the HET $N_2O_5$ pathway at night), which potentially lead to underestimate in the nocturnal pathway contribution to $NO_3^-$.

Minor Comments:

1. Line 90 and 132-133: Check grammar.

Responses: Corrected. We have carefully revised the manuscript and corrected the grammatical sentences in Lines 88-91 and 132-135.

2. Line 102-104 and 133-134: Rephrase the sentence to improve clarity.

Responses: Corrected. We have rephrased the sentences, as following in Lines 102-104: "Prabhakar et al. (2017) revealed that the active nocturnal $NO_3^-$ formation from the upper PBL contributed 80 % to daytime surface $NO_3^-$ concentrations in winter of 2013 in California.", and lines 133-135: "Most previous studies have focused on only a few short period of $NO_3^-$ pollution episodes, and lacked the seasonal analysis for the full year.".

3. Fig S4, S8, S10 and Table S7 have not been referred to in the main manuscript.

Responses: Corrected. Table S7 and Figs S4, S8, S10 have been removed in the Supplementary material. The manuscript has been revised accordingly.

**RC3, Reviewer #3:**

General Comments

This study quantified the contributions of local emissions and transport from other regions both directly and indirectly to the particulate and total nitrate formation by comparing the results of CMAQ simulation with different emission scenarios. The study also suggests the contributions of chemical formation pathways, aerosol processes, physical transports, and dry deposition to the total and particulate nitrate formation using the CMAQ process analysis tool (Integrated Process Rate (IPR) and Integrated Reaction Rate (IRR)). Since the proportion of nitrate in $PM_{2.5}$ has been increasing in recent years, understanding the mechanisms of particulate and total nitrate formation is important. In this respect, this study suggests interesting and meaningful scientific information. Considerable indirect transport effects on nitrate formation are of particular interest.

Nonetheless, this reviewer suggests minor revisions to the manuscript and raises a few questions or concerns that would hopefully be addressed in the revised manuscript to improve readers' understanding.

Major Comments

1. The authors used IPR and IRR to quantify the contributions of various processes to the formation of particulate nitrate, gaseous HNO3, and total nitrate, and these are one of the main results of this study. However, insufficient information on IPR and IRR can make readers confused who are not familiar with CMAQ models. IRR includes chemical reactions in the aerosol phase (which is included in AERO), correct?

Responses: In the manuscript, the Integrated Process Rate (IPR) was applied to investigate the cumulative effect of chemical and physical processes to $NO_3^-$ formation within the PBL. In Figure 5(a, c, e, g), AERO processes (including condensation, coagulation, and aerosol growth) are the dominant contributors of $NO_3^-$ formation. Furthermore, the Integrated Reaction Rate (IRR) analysis was employed to quantify the rates of $TNO_3$ (sum of $NO_3^-$ and $HNO_3$) chemical reactions pathways. Figure 6(b) illustrates that $TNO_3$ chemical production is dominated by the $OH+NO_2$ pathway on the daily timescale, accounting for 69.3–86.9 % in Shanghai. Conceptually, AERO processes is different from chemical reaction pathways.

Minor Comments

1. Line 87: dinitrogen àdinitrogen pentoxide?

Responses: We have corrected the manuscript, as following in Line 87: "the heterogeneous hydrolysis of dinitrogen àdinitrogen pentoxide (HET $N_2O_5$) on the wet particles' surface".

2. Line 163 – 164: "Detailed configurations of the WRF model provided in the studies of Hu et al. (2016) and Wang et al. (2021), shown in Table S1,." --> not easy to understand.

Responses: We have corrected the manuscript, see Lines 164-165: "The detailed configurations of the WRF model shown in Table S1, consistent with Hu et al. (2016) and Wang et al. (2021)".

3. Line 199: Unlike $F_{Local}$, $F_{Direct}$, $F_{Background}$, $F_{Indirect}$ is not easily understood. Suggest adding a brief explanation for this term.

Responses: Thanks for the advice. For the YRD as the target region, indirect transport contribution ($F_{indirect}$) represents that $NO_3^-$ generated from chemical reactions between outside-transported and local-emitted precursors, as suggested by (Qu et al., 2021). $F_{Region}$, $F_{Direct}$, and $F_{Indirect}$ were defined in Eq. (2-4), respectively.

$$F_{Region} = (C_{base} - C_{outside-zero})/C_{base} \qquad (2)$$

$$F_{Direct} = (C_{YRD-zero} - C_{all-zero})/C_{base} \qquad (3)$$

$$F_{Indirect} = [(C_{base} - C_{outside-zero}) - (C_{YRD-zero} - C_{all-zero})]/C_{base} \qquad (4)$$

in which ($C_{base}$ - $C_{outside-zero}$) represents is the sum concentrations of indirect and direct transport (the total regional transport), and ($C_{YRD-zero}$ - $C_{all-zero}$) quantifies the direct transport concentrations. Regional transport ($F_{Region}$) represents the sum of direct and indirect transport contribution from outside regions.

We have added above explanation and in the revised manuscript in Lines 193-202.

4. Line 273: "The results in this study show a better model performance" à Quantitatively, how much the model performance in this study is better than previous studies? What are the most significant differences between this and previous models that affected the model performance?

Responses: It should be emphasized that the description of "Compared to past studies" is unaccurate. As shown in Table 2 and Figure S1, the daily time series of prediction and observation PM2.5, NO2, and O3 concentrations were well-captured in the four seasons of 2017 in selected cities. One possible reason is that the WRF performance is better than our previous works. Other reasons could be associated with using the local anthropogenic emission inventory (SAES, with high-resolution and based on local measure data) to simulate the 2017 YRD region. Hence, the seasonal results of statistical metrics show a better model performance than our previous works. We have checked the description of this sentence. The manuscript has been revised as following in Lines 274-276:

"When compared to our previous studies (Hu et al., 2016;Wang et al., 2021;Ma et al., 2021;Sulaymon et al., 2021;Li et al., 2021), the statistical results in this study show a better model performance."

5. Line 274–276 and 283–286: In Fig. S1, predicted $PM_{2.5}$ appears to agree quite well with the observed time series but predicted nitrate in Fig. 2 was underestimated, particularly in April and October. Isn't nitrate an important

species for those periods? or overestimation of other species compensated for these underestimations? Or underestimates of RH can explain this?

Responses: The insufficient heterogeneous chemistry on the dust particles' surface and uncertainties in precursors emissions in the CMAQ model affect the model performance of $NO_3^-$ during the spring and autumn (Xie et al., 2022). We have added above explanation and in the revised manuscript in Lines 472-474.

6. Line 299: year respectively à year, respectively.

Responses: We have revised the manuscript, as following in Lines 304-305: "The $NO_3^-$ decreased by 12.0, 6.9, 0.9, 4.8 and 6.1 μg m$^{-3}$ in winter, spring, summer, autumn, and a year, respectively,".

7. Line 360: Vertical mixing would be acceptable. However, the development of the PBL starts after sunrise, but the authors discuss the nighttime process.

Responses: Corrected. The description of "through the vertical mixing and development of the PBL" is unaccurate. During midnight (the lowest height of PBL), the nocturnal stable PBL weakens the vertical transport and accelerates the pollutant accumulation near the surface. We have checked the description of this sentence. The manuscript has been revised as following in Lines 364-366:
 "TRAN processes constitute the dominant source during midnight (1:00–6:00 am), owing to the stable PBL weakening the contribution of vertical transport and accelerating the accumulation of $NO_3^-$ concentrations at the surface."

8. Line 380: To avoid confusion, "opposite with the first-peak time of NO3- production".

Responses: Thanks for the kind advice. We have carefully revised the manuscript and corrected this sentences (see Lines 385-386): "However, in the winter, the peak times of $HNO_3$ production are opposite with the first-peak time of $NO_3^-$ production, but consistent with the second-peak time".

9. Line 404: Incorrect use of "respectively."

Responses: We have corrected the manuscript, as following in Lines 411-413: "The averaged $TNO_3$ production rates are 0.31 ±0.14, 0.65 ±0.37, 1.09 ±0.68, and 0.28 ± 0.22 ppb h$^{-1}$ in the winter, spring, summer, and autumn, respectively (Table S6).".

10. Line 450–451 and 454–456: TRAN (vertical and horizontal transport) processes were the largest SINK in particulate nitrate formation. In Line 450–451, regional transport (although only direct transport is considered) accounts for about 15%. How does "sink" contribute to nitration formation? Is TRAN different from regional transport conceptually?

Responses: This study investigated the contributions of local emission and regional transport to $NO_3^-$ concentrations in the YRD, as well as the dominant processes and chemical pathways to $NO_3^-$ production in Shanghai. Local emissions dominate $NO_3^-$ concentrations in the YRD (50–62%), while regional transport contributes 38–50% to

NO$_3^-$ concentrations. TRAN processes act as the main negitive contributor to NO$_3^-$ production within the PBL in Shanghai. Conceptually, TRAN processes is different from regional transport.

References:

An, J., Huang, Y., Huang, C., Wang, X., Yan, R., Wang, Q., Wang, H., Jing, S., Zhang, Y., Liu, Y., Chen, Y., Xu, C., Qiao, L., Zhou, M., Zhu, S., Hu, Q., Lu, J., and Chen, C.: Emission inventory of air pollutants and chemical speciation for specific anthropogenic sources based on local measurements in the Yangtze River Delta region, China, Atmos. Chem. Phys., 21, 2003-2025, 10.5194/acp-21-2003-2021, 2021.

Li, L., Xie, F., Li, J., Gong, K., Xie, X., Qin, Y., Qin, M., and Hu, J.: Diagnostic analysis of regional ozone pollution in Yangtze River Delta, China: A case study in summer 2020, Sci. Total Environ., 151511, https://doi.org/10.1016/j.scitotenv.2021.151511, 2021.

Liu, T., Wang, X. Y., Hu, J. L., Wang, Q., An, J., Gong, K., Sun, J., Li, L., Qin, M., Li, J., Tian, J., Huang, Y., Liao, H., Zhou, M., Hu, Q., Yan, R., Wang, H., and Huang, C.: Driving Forces of Changes in Air Quality during the COVID-19 Lockdown Period in the Yangtze River Delta Region, China, Environ. Sci. Technol. Lett., 7, 779-786, 10.1021/acs.estlett.0c00511, 2020.

Qin, M., Hu, A., Mao, J., Li, X., Sheng, L., Sun, J., Li, J., Wang, X., Zhang, Y., and Hu, J.: PM2.5 and O3 relationships affected by the atmospheric oxidizing capacity in the Yangtze River Delta, China, Sci. Total Environ., 152268, https://doi.org/10.1016/j.scitotenv.2021.152268, 2021.

Qu, K., Wang, X. S., Xiao, T., Shen, J., Lin, T., Chen, D., He, L.-Y., Huang, X.-F., Zeng, L., Lu, K., Ou, Y., and Zhang, Y.: Cross-regional transport of PM2.5 nitrate in the Pearl River Delta, China: Contributions and mechanisms, Sci. Total Environ., 753, 142439, https://doi.org/10.1016/j.scitotenv.2020.142439, 2021.

Xie, X., Hu, J., Qin, M., Guo, S., Hu, M., Wang, H., Lou, S., Li, J., Sun, J., Li, X., Sheng, L., Zhu, J., Chen, G., Yin, J., Fu, W., Huang, C., and Zhang, Y.: Modeling particulate nitrate in China: current findings and future directions, Environ. Int., 107369, https://doi.org/10.1016/j.envint.2022.107369, 2022.